# Are Tabular Foundation Model Rankings Reliable?
# A Generalizability Theory Analysis of RelBench and DBInfer

**Dinesh Katupputhur Ramprasath** [1]  **Tom Palczewski** [1]  **Joe Meyer** [1]  **Roshan Reddy Upendra** [1]  **Minghua Li** [1]

## Abstract

Tabular and relational learning foundation models are often evaluated on multi-task relational benchmarks such as RelBench and DBInfer, where models are ranked by aggregated task-level performance. However, how *reliable* are these rankings? We apply Generalizability Theory (G-theory), a variance decomposition framework from measurement science, to quantify ranking reliability across 14 datasets (9 RelBench + 5 DBInfer), 48 tasks, and up to 35 models from 11 families. We decompose score variance into model differences (signal), task effects, model×task interaction, and sampling error. Our analysis yields three main findings: (1) only 2 of 14 datasets achieve $E\rho^2 > 0.80$, the psychometric threshold for reliable measurement; (2) Decision-study (D-study) simulations show that 80–99% of test items can be removed while maintaining ranking stability ($\rho > 0.90$), revealing massive redundancy; (3) Model×task interaction, which indicates task-dependent model behavior, is substantial across several datasets and dominates in some cases, reaching 81.2% of variance on rel-stack. Collectively, these findings suggest that single-number leaderboard rankings of tabular and relational learning foundation models may provide an unstable estimate of relative model performance.

## 1. Introduction

Tabular foundation models (TFMs) and relational learning foundation models (RFMs) such as TabPFN (Hollmann et al., 2023), TabICL (Ye et al., 2024), ContextTab (Marco Spinaci, 2025), Griffin (Yanbo Wang, 2025), GNN+TabPFN-2.5 (Joe Meyer & Palczewski, 2026), RT (Rishabh Ranjan, 2025), KumoRFM (Matthias Fey, 2025; Valter Hudovernik, 2026) have recently challenged narrow machine learning models, including gradient-boosted decision trees (Friedman, 2001) and auto ML approaches like autogluon (Nick Erickson, 2020), on relational benchmarks like RelBench (Robinson et al., 2024) and DBInfer (Fey et al., 2023). These benchmarks aggregate task-level performance across multiple prediction tasks and rank models on a leaderboard.

As the structured data foundation model landscape evolves rapidly, with recent scaling advances such as TabPFN v2 (Müller et al., 2025), v2.5 (Léo Grinsztajn, 2026), and TabICL v2 (Jingang Qu, 2026) substantially extending model capabilities, it becomes scientifically important to understand whether benchmark rankings reflect genuine model differences or are artifacts of evaluation design. Reliable measurement is a prerequisite for drawing valid scientific conclusions about model capabilities and guiding principled model development.

However, the reliability of such rankings remains unclear. If dropping one task, or changing the test split, reshuffles the leaderboard, the ranking captures task-dependent model behavior rather than stable model capability. Despite a growing literature questioning ML benchmark validity, including Item Response Theory (IRT)-based item analysis (Martínez-Plumed et al., 2019; Rodriguez et al., 2021; Vania et al., 2021), benchmark compression (Polo et al., 2024), Bayesian classifier comparisons (Benavoli et al., 2017), and ranking-based generalizability frameworks (Matteucci et al., 2024), no prior work has quantified the reliability of structured data foundation model rankings using a formal measurement-theoretic framework.

Very recently, Messing (2026) applied Generalizability Theory (G-theory) (Cronbach et al., 1972; Brennan, 2001) to decompose measurement error in LLM evaluation pipelines (annotation, safety, MMLU), showing that prompt phrasing and judge model choice inject substantial hidden variance. While this work is a preprint, G-theory itself is a well-established framework in psychometrics with decades of peer-reviewed validation (Cronbach et al., 1972; Brennan, 2001); Messing's contribution is demonstrating its applicability to AI evaluation, which we build upon. Their focus is

[1]SAP, Palo Alto, CA, USA. Correspondence to: Dinesh Katupputhur Ramprasath <dinesh.katupputhur.ramprasath@sap.com>, Tom Palczewski <tom.palczewski@sap.com>.

*Proceedings of the $2^{nd}$ ICML Workshop on Foundation Models for Structured Data*, Seoul, South Korea. 2026. Copyright 2026 by the author(s).

on LLM-as-judge pipelines with prompt/temperature facets, whereas our setting, multi-task relational benchmarks with deterministic model predictions, involves fundamentally different variance sources: task heterogeneity and model×task interaction. The mathematical framework (two-way random-effects ANOVA with variance decomposition) is identical and well-established; what differs is the facet structure.

We apply G-theory (Brennan, 2001; Cronbach et al., 1972), a variance decomposition framework from psychometrics that extends classical test theory (Lord & Novick, 1968), to quantify where score variability comes from. G-theory decomposes observed score variance into: (1) $\sigma_M^2$: variance due to models (the signal we want); (2) $\sigma_T^2$: variance due to tasks (irrelevant to ranking); (3) $\sigma_{MT}^2$: model×task interaction (rankings change depending on which tasks are included); (4) $\sigma_e^2$: sampling error from finite test sets.

The generalizability coefficient can be defined as follows $E\rho^2 = \sigma_m^2/(\sigma_m^2 + \sigma_\delta^2)$ quantifies ranking reliability, where $\sigma_\delta^2$ is the relative error variance and $\sigma_m^2$ is the universe-score variance (true differences among measurement objects). In our case, the generalizability coefficient can be defined as in Equation 5.

Across 14 datasets (9 RelBench + 5 DBInfer), 48 tasks, and up to 35 models: (1) only 2 of 14 datasets achieve $E\rho^2 > 0.80$, the psychometric standard for reliable ranking (Brennan, 2001); (2) model×task interaction ($\sigma_{MT}^2$) is the largest variance component on most datasets, reaching 81.2% of total variance on rel-stack, meaning the leaderboard ranking depends critically on which tasks happen to be included; and (3) Decision-study (D-study) simulations reveal 80–99% item redundancy on several datasets, while showing that task diversity, not test set size, is the binding constraint on reliability.

## 2. Method

### 2.1. Data and Models

We analyze 14 datasets from RelBench (9 datasets: rel-f1, rel-amazon, rel-salt, rel-stack, rel-hm, rel-event, rel-avito, rel-arxiv, rel-ratebeer) and DBInfer (5 datasets: dbinfer-amazon, dbinfer-seznam, dbinfer-stackexchange, dbinfer-diginetica, dbinfer-avs). Each dataset contains 1–8 prediction tasks (48 total), evaluated on up to 35 models from 11 families: XGBoost variants (4), gradient boosting (Light-GBM, CatBoost, etc., 5), tree ensembles (2), linear/kernel (4), GNN (1), tabular FMs (TabPFN, TabICL, Context-Tab, 3), RDBLearn+DFS (3), AutoGluon (1), deep tabular (RealMLP, TabSTAR, 2), and Dummy baseline (1).

Why include single-table models on multi-table benchmarks? Tabular FMs (TabPFN, TabICL, ContextTab) are single-table learners that cannot directly consume relational

databases. We include them for two reasons: (1) they represent the current state-of-the-art for tabular prediction and serve as strong baselines against which any relational method must be compared; (2) by evaluating them both standalone (on flat-table features via simple joins) and wrapped by RDBLearn (which adds automatically synthesized relational features via Deep Feature Synthesis (Zhang et al., 2025)), we can isolate the contribution of relational structure. Similarly, classical models (XGBoost, LightGBM, etc.) receive the same flat-table features. GNN operates natively on the multi-table relational graph.

For each model×task combination, we have per-instance binary predictions (correct/incorrect), from which we compute the cell mean (accuracy per model per task).

### 2.2. Variance Decomposition

We treat the cell means $\bar{Y}_{mt}$ (accuracy of model $m$ on task $t$) as observations in a two-way random-effects ANOVA:

$$\bar{Y}_{mt} = \mu + \alpha_m + \beta_t + (\alpha\beta)_{mt} + \varepsilon_{mt} \tag{1}$$

where $\alpha_m \sim (0, \sigma_M^2)$, $\beta_t \sim (0, \sigma_T^2)$, $(\alpha\beta)_{mt} \sim (0, \sigma_{MT}^2)$, and $\varepsilon_{mt}$ captures binomial sampling error with known variance $\sigma_e^2 = p(1-p)/n_I$. Variance components are estimated via Method of Moments from mean squares:

$$\hat{\sigma}_{MT}^2 = \max(0, MS_{MT} - \hat{\sigma}_e^2) \tag{2}$$

$$\hat{\sigma}_M^2 = \max(0, (MS_M - MS_{MT})/n_T) \tag{3}$$

$$\hat{\sigma}_T^2 = \max(0, (MS_T - MS_{MT})/n_M) \tag{4}$$

The generalizability coefficient for relative decisions (ranking) with $n_T$ tasks is:

$$E\rho^2 = \frac{\sigma_M^2}{\sigma_M^2 + \sigma_{MT}^2/n_T + \sigma_e^2/n_T} \tag{5}$$

The dependability coefficient for absolute decisions is:

$$\Phi = \frac{\sigma_M^2}{\sigma_M^2 + \sigma_{MT}^2/n_T + \sigma_T^2/n_T + \sigma_e^2/n_T} \tag{6}$$

*Applicability to ML benchmarks.* G-theory is a general variance decomposition framework, not specific to educational testing. Its only structural requirement is that observations can be crossed by facets (here: models × tasks). Benchmark tasks are curated rather than randomly sampled, but this is standard in G-theory applications; raters in performance assessment are also not randomly sampled (Brennan, 2001). The interpretation is: how stable are rankings across the *observed* set of tasks?

### 2.3. D-Study Simulations

We assess benchmark efficiency through Decision studies (D-studies), a standard G-theory tool for projecting relia-

bility under different measurement designs: *Item subsampling.* For each fraction $f \in \{0.01, \ldots, 1.0\}$, we randomly subsample $f$ of items within each task, recompute model rankings, and measure Spearman $\rho$ with the full-data ranking (30 repetitions); *Task subsampling.* We subsample $k \in \{1, \ldots, n_T\}$ tasks and measure ranking stability; *Leave-one-task-out instability.* For each model, we compute the standard deviation and range of its rank across 50 leave-one-task-out trials;

## 3. Results

### 3.1. Variance Decomposition

Table 1 presents the variance decomposition for all 14 datasets.

*Table 1.* Variance decomposition and generalizability coefficients. %$M$: proportion of variance from model differences (signal). %$T$: proportion of varaince from task differences, %$MT$: model×task interaction (task-dependent ranking). %$e$: proportion of variance from sampling error. $E\rho^2$ is the generalizability coefficient for relative decisions (ranking). $\Phi$ is dependability coefficient for absolute decisions. Bold: $E\rho^2 < 0.80$. Note: single-task datasets (dbi-diginetica, dbi-avs) cannot separate $\sigma^2_{MT}$ from $\sigma^2_e$; their $E\rho^2$ reflects within-task reliability only.

| Dataset | $M$ | $T$ | %$M$ | %$T$ | %$MT$ | %$e$ | $E\rho^2$ | $\Phi$ |
|---|---|---|---|---|---|---|---|---|
| *RelBench* | | | | | | | | |
| rel-salt | 34 | 8 | 12.4 | 61.0 | 26.6 | 0.0 | **0.788** | 0.531 |
| rel-amazon | 31 | 5 | 1.0 | 91.2 | 7.8 | 0.0 | **0.394** | 0.049 |
| rel-f1 | 35 | 3 | 0.3 | 98.6 | 0.8 | 0.4 | **0.393** | 0.008 |
| rel-ratebeer | 33 | 5 | 1.8 | 81.8 | 16.4 | 0.1 | **0.355** | 0.084 |
| rel-arxiv | 35 | 3 | 2.2 | 83.2 | 14.5 | 0.1 | **0.315** | 0.064 |
| rel-hm | 31 | 3 | 1.6 | 87.3 | 11.1 | 0.0 | **0.302** | 0.047 |
| rel-event | 35 | 3 | 1.4 | 79.0 | 19.1 | 0.5 | **0.176** | 0.041 |
| rel-stack | 34 | 3 | 0.3 | 18.4 | 81.2 | 0.0 | **0.011** | 0.009 |
| rel-avito | 34 | 3 | 0.0 | 44.5 | 55.4 | 0.1 | **0.000** | 0.000 |
| *DBInfer* | | | | | | | | |
| dbi-seznam | 29 | 2 | 95.3 | 0.5 | 4.2 | 0.0 | 0.979 | 0.976 |
| dbi-diginetica | 29 | 1 | 95.3 | — | — | 4.7 | 0.953 | 0.953 |
| dbi-avs | 32 | 1 | 21.0 | — | — | 79.0 | **0.210** | 0.210 |
| dbi-stackex. | 29 | 2 | 1.1 | 86.9 | 12.0 | 0.0 | **0.150** | 0.021 |
| dbi-amazon | 32 | 2 | 0.0 | 68.9 | 31.0 | 0.1 | **0.000** | 0.000 |

Finding 1: Single-number aggregate rankings are often unreliable. Only 2 of 14 datasets (dbinfer-seznam, dbinfer-diginetica) achieve $E\rho^2 > 0.80$. A third (rel-salt) approaches acceptability at $E\rho^2 = 0.788$. The remaining 11 datasets fall below 0.40, meaning more than 60% of ranking variance reflects task-dependent model behavior rather than stable model differences.

Finding 2: Model×task interaction is important in some cases. On rel-stack, 81.2% of variance comes from $\sigma^2_{MT}$, indicating that rankings depend entirely on which tasks are included. On rel-avito ($\sigma^2_{MT} = 55.4\%$) and dbinfer-amazon (31.0%), the pattern is similar. A model ranked #1 on one task subset may rank in the bottom third on another, reflecting limited cross-task generalization rather than random error.

Finding 3: Task variance dominates on most datasets. On

rel-f1, 98.6% of variance comes from task differences ($\sigma^2_T$), with only 0.3% from model differences. This means task selection matters far more than model selection.

### 3.2. Ranking Instability

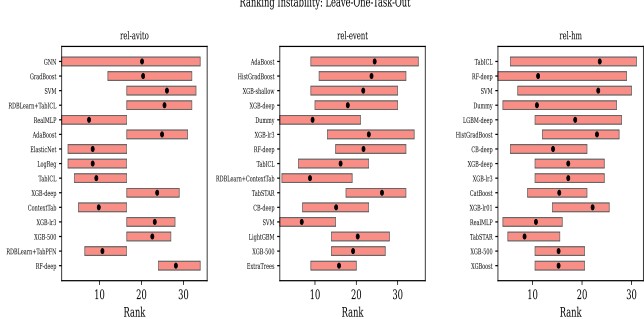

*Figure 1.* Leave-one-task-out ranking instability. Red bars show the rank range (short bars = stable rankings; long bars = rank changes substantially when one task is removed); black dots show mean rank. On rel-avito, GNN's rank ranges from 1 to 34 depending on which task is dropped.

Figure 1 visualizes ranking instability under leave-one-task-out perturbation.

Key observations: (1) GNN on rel-avito: rank ranges from 1 to 34 ($\sigma_{\text{rank}} = 16.3$). GNN ranks #1 on user-visits but last on ad-ctr, making its aggregate rank uninformative. (2) TabICL on rel-arxiv: rank ranges over 22 positions. On paper-citation, TabICL excels; on author-category, it underperforms. (3) AdaBoost on rel-event: rank ranges over 26 positions, driven by task-specific class imbalance.

These examples span distinct model families (graph-based, foundation model, classical ensemble) and illustrate that ranking instability is pervasive, not limited to any single model type.

### 3.3. D-Study: Benchmark Efficiency

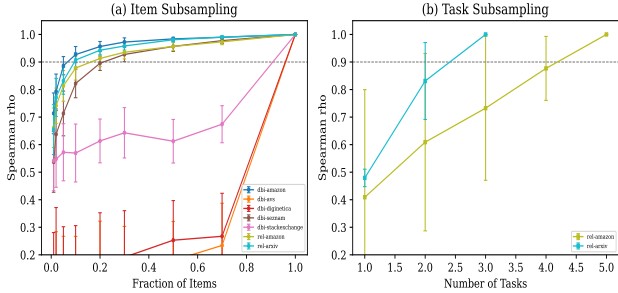

*Figure 2.* D-study simulations. (a) Item subsampling: most datasets reach $\rho \geq 0.90$ with 5–20% of items. (b) Task subsampling: adding tasks helps but saturates quickly.

D-study results (Figure 2) reveal that (1) rel-salt: only 1% of items needed ($\rho \geq 0.90$ with 800 of 80,000 items), indicating 99% redundancy,(2) rel-event: 5% sufficient (208 of

4,162),(3) rel-amazon: 20% sufficient (10,000 of 50,000),(4) rel-avito: requires 70% of items, reflecting genuine difficulty spread.

Task subsampling shows steeper degradation: dropping from 3 to 2 tasks typically reduces $\rho$ by 0.1–0.3, confirming that task diversity is the binding constraint, not item count.

## 4. Discussion and Recommendations

Prior work on benchmark reliability operates at different granularities. At the item level, IRT-based analyses (Martínez-Plumed et al., 2019; Rodriguez et al., 2021; Vania et al., 2021) quantify how informative individual test instances are for discriminating between models. Polo et al. (2024) leveraged this to compress LLM benchmarks by 95%. At the pipeline level, Messing (2026) used G-theory to decompose variance from prompt phrasing, temperature, and judge model in LLM-as-judge evaluations. At the study level, Matteucci et al. (2024) proposed a ranking-based framework using Maximum Mean Discrepancy to measure whether experimental conclusions generalize across datasets. Our work fills a gap at the task level: we decompose variance in multi-task benchmarks into model signal vs. task effects vs. model×task interaction, directly quantifying whether aggregate leaderboard rankings are trustworthy.

G-theory (Cronbach et al., 1972; Brennan, 2001) extends classical test theory (Lord & Novick, 1968) by partitioning variance across multiple facets. While standard in educational measurement, best to our knowledge, it has not previously been applied to ML benchmark evaluation for tabular models.

Our G-theory analysis reveals that the vast majority of Rel-Bench (Robinson et al., 2024; Fey et al., 2023) and DBInfer benchmarks produce rankings with limited cross-task stability. Model×task interaction ($\sigma^2_{MT}$) accounts for a substantial portion of the variance on several tasks ranging up to 81.2% on rel-stack. This interaction reflects genuine task-dependent model behavior: different models have different strengths on different tasks, and a single aggregate number cannot capture this structure.

These results do not directly show whether individual structured data foundation models generalize across all task types. Rather, they show that aggregate benchmark rankings may be unstable or difficult to interpret when task effects and model×task interactions are large. Future evaluations of FMs should therefore report task-level behavior alongside aggregate leaderboard ranks. The rank instability observed for AutoGluon and RDBLearn suggests that ensemble and feature-engineering approaches may face similar task-composition sensitivity.

Practical implications: (1) Report $E\rho^2$ alongside accuracy.

Any dataset with $E\rho^2 < 0.80$ should carry a reliability warning, alerting users that the ranking may not be stable across task subsets. (2) Increase task diversity. High model×task interaction means that rankings depend on the task mix. Adding more items on existing tasks does not improve reliability; adding diverse tasks does ($E\rho^2$ scales with $n_T$, not $n_I$). (4) Report per-task rankings. Aggregate rankings mask task-specific behavior (e.g., GNN on rel-avito: rank 1–34). Reporting per-task or task-group rankings enables users to select models appropriate for their specific use case.

Limitations: Our analysis treats tasks as exchangeable random effects, which may overestimate $\sigma^2_{MT}$ when tasks are intentionally heterogeneous. Single-task datasets (dbinfer-avs, dbinfer-diginetica) cannot separate model×task interaction from sampling variance; their $E\rho^2$ values reflect within-task reliability only and should not be interpreted as evidence of cross-task stability.

## 5. Conclusion

To our knowledge, we present the first Generalizability Theory analysis of tabular foundation and relational learning model benchmarks. Only 2 of 14 datasets achieve $E\rho^2 > 0.80$, the psychometric standard for reliable ranking. Model×task interaction can be an important variance component for some datasets, meaning leaderboard rankings are contingent on task selection. These findings call for reliability-aware benchmark design: more diverse tasks, fewer redundant items, and reporting of generalizability coefficients. As FMs continue to scale, with recent advances extending model capabilities to larger datasets and broader task types, G-theory provides a principled framework for evaluating whether improved aggregate performance reflects genuine cross-task generalization.

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

# A. Experimental Setup: Parameters, Libraries, and Settings

This section documents all software, parameters, and computational settings used to produce the results in this paper, to support reproducibility.

## A.1. Software and Libraries

All analyses were implemented in Python 3.8. Table 2 lists the key libraries and their versions.

*Table 2.* Software dependencies and versions.

| Library | Version / Notes |
|---|---|
| Python | 3.8 (Azure ML environment) |
| NumPy | 1.24.x |
| Pandas | 2.0.x |
| SciPy | 1.10.x (`scipy.stats.spearmanr`) |
| Matplotlib | 3.7.x (`Agg` backend) |
| `torch_measure` | Stanford CS321M package |
| PyTorch | 2.1.x (IRT fitting) |

## A.2. Data Source and Preprocessing

**Data source.** Model evaluation results were obtained from an exhaustive evaluation pipeline on the RelBench (Fey et al., 2023) and DBInfer benchmark suites, stored in `all_merged.pkl`. This file contains per-item binary responses for up to 39 models across 17 datasets and 55 tasks. The 14 datasets reported in this paper are those with $\geq 10$ models per task and $\geq 3$ common models across tasks.

**Binarization.** All responses are binary: $Y_{ij} \in \{0, 1\}$, where 1 indicates the model's prediction matched the ground truth. For regression tasks, predictions were binarized at the per-task median.

**Item cap.** To keep computation tractable and prevent large-item datasets from dominating, the number of items per task was capped at $n_I^{\max} = 10,000$. Tasks with fewer items used all available items.

**Minimum model threshold.** Tasks with fewer than `MIN_MODELS_PER_TASK = 10` models were excluded from analysis. For multi-task datasets, only models appearing in *all* tasks for that dataset were retained to enable balanced G-theory designs.

## A.3. G-Theory Variance Decomposition

We use a two-facet crossed design: **Model** $(M) \times$ **Task** $(T)$, with items nested within tasks.

**Variance components.** Mean squares are computed from the accuracy pivot table $Y_{mt}$ (model $\times$ task cell means):

$$\text{MS}_M = n_T \sum_m (\bar{Y}_{m\cdot} - \bar{Y}_{\cdot\cdot})^2 / (n_M - 1) \tag{7}$$

$$\text{MS}_T = n_M \sum_t (\bar{Y}_{\cdot t} - \bar{Y}_{\cdot\cdot})^2 / (n_T - 1) \tag{8}$$

$$\text{MS}_{MT} = \sum_{m,t} (Y_{mt} - \bar{Y}_{m\cdot} - \bar{Y}_{\cdot t} + \bar{Y}_{\cdot\cdot})^2 / \text{df}_{MT} \tag{9}$$

Sampling variance $\sigma_e^2$ is estimated as the mean of $\hat{p}_{mt}(1 - \hat{p}_{mt})/n_{mt}$ across all cells, where $\hat{p}_{mt}$ is the cell accuracy and $n_{mt}$ is the number of items.

Variance components are then:

$$\hat{\sigma}_{MT}^2 = \text{MS}_{MT} - \hat{\sigma}_e^2 \tag{10}$$

$$\hat{\sigma}_M^2 = (\text{MS}_M - \text{MS}_{MT})/n_T \tag{11}$$

$$\hat{\sigma}_T^2 = (\text{MS}_T - \text{MS}_{MT})/n_M \tag{12}$$

Negative estimates are set to zero.

**Generalizability coefficients.**

$$\sigma_{\text{rel}}^2 = \hat{\sigma}_{MT}^2/n_T + \hat{\sigma}_e^2/n_T \tag{13}$$

$$E\rho^2 = \hat{\sigma}_M^2/(\hat{\sigma}_M^2 + \sigma_{\text{rel}}^2) \tag{14}$$

$$\sigma_{\text{abs}}^2 = \hat{\sigma}_{MT}^2/n_T + \hat{\sigma}_T^2/n_T + \hat{\sigma}_e^2/n_T \tag{15}$$

$$\Phi = \hat{\sigma}_M^2/(\hat{\sigma}_M^2 + \sigma_{\text{abs}}^2) \tag{16}$$

## A.4. D-Study Parameters

**Item subsampling.** For each dataset, we drew item subsets at fractions $f \in \{0.01, 0.02, 0.05, 0.1, 0.2, 0.3, 0.5, 0.7, 1.0\}$ of the total item count. At each fraction, 30 independent random draws were performed (without replacement within each draw). Model rankings were computed from the subsampled accuracy and compared to the full-data ranking via Spearman $\rho$.

**Task subsampling.** For datasets with $n_T \geq 3$ tasks, we varied $k$ from 1 to $n_T$, sampling $k$ tasks uniformly at random (30 repetitions per $k$). Accuracy was averaged across the sampled tasks and ranked; Spearman $\rho$ was computed against the full-task ranking.

**Ranking instability.** Leave-one-task-out analysis with 50 repetitions per dataset. In each repetition, one task was dropped at random, model rankings were recomputed, and rank statistics (mean, std, min, max, range) were recorded per model.

## A.5. Computational Environment

All G-theory analyses were run on a single Azure ML compute instance (Standard_NC24ads_A100_v4, NVIDIA A100

40GB GPU, 24 vCPUs, 220GB RAM). The full analysis pipeline (14 datasets, all D-studies, all figures) completes in approximately 5 minutes. Random seed: `SEED = 42` for all stochastic operations.

### A.6. Figure Generation

All figures use Matplotlib with `font.size=11`, `figure.dpi=200`, `font.family=serif`, and `savefig.bbox=tight`. Output format: PDF (vector) for paper figures, PNG (raster) for supplementary inspection. Color palette: Material Design colors (`#2196F3` for $\sigma_M^2$, `#FF9800` for $\sigma_T^2$, `#f44336` for $\sigma_{MT}^2$, `#9E9E9E` for $\sigma_e^2$). The `tab10` colormap is used for D-study line plots.

### A.7. Model Hyperparameters

Tables 3 to 5 list the complete hyperparameter settings for all 39 models evaluated. All models use a fixed random seed $S = 42$.

### A.8. IRT Fitting (for companion analysis)

Where IRT results are referenced, parameters are as follows. Rasch and 2PL models: 200 epochs, learning rate $= 0.05$, weight decay $= 0.01$, optimized via Adam. LogisticFM: factor dimensions $K \in \{1, 2, 4\}$. Bootstrap confidence intervals: 100 resamples with 80/20 random train/test masks. All IRT fitting uses the `torch_measure` GPU-accelerated package.

## B. Full Variance Decomposition Results

Table 6 provides the complete variance decomposition including raw variance components ($\sigma^2$), mean squares (MS), standard error of measurement (SEM), and the average number of test items per task.

## C. D-Study: Item Subsampling

Table 7 shows Spearman $\rho$ between full-data and subsampled rankings as a function of item fraction (30 repetitions per fraction).

## D. D-Study: Task Subsampling

Table 8 shows ranking stability as a function of the number of tasks included (datasets with $\geq 3$ tasks only).

*Table 3.* Sklearn model hyperparameters. Training set capped at 200,000 samples. Default parameters unless noted.

| Model | Library | Key Hyperparameters |
|---|---|---|
| XGBoost | xgboost | n_estimators=100, max_depth=6, lr=0.1, eval_metric=logloss |
| XGB-deep | xgboost | n_estimators=200, max_depth=10, lr=0.1 |
| XGB-shallow | xgboost | n_estimators=50, max_depth=3, lr=0.1 |
| XGB-lr01 | xgboost | n_estimators=200, max_depth=6, lr=0.01 |
| XGB-lr3 | xgboost | n_estimators=100, max_depth=6, lr=0.3 |
| XGB-d4 | xgboost | n_estimators=150, max_depth=4, lr=0.1 |
| XGB-500 | xgboost | n_estimators=500, max_depth=6, lr=0.05 |
| LightGBM | lightgbm | n_estimators=100, max_depth=6, lr=0.1 |
| LGBM-deep | lightgbm | n_estimators=200, max_depth=12, lr=0.1 |
| LGBM-lr01 | lightgbm | n_estimators=200, max_depth=6, lr=0.01 |
| CatBoost | catboost | iterations=200, depth=6, lr=0.1, task_type=GPU |
| CB-deep | catboost | iterations=200, depth=10, lr=0.1, task_type=GPU |
| CB-lr01 | catboost | iterations=300, depth=6, lr=0.01, task_type=GPU |
| RandomForest | sklearn | n_estimators=200, max_depth=10, n_jobs=-1 |
| RF-deep | sklearn | n_estimators=200, max_depth=20, n_jobs=-1 |
| RF-shallow | sklearn | n_estimators=200, max_depth=5, n_jobs=-1 |
| ExtraTrees | sklearn | n_estimators=200, max_depth=10, n_jobs=-1 |
| HistGradBoost | sklearn | max_iter=200, max_depth=6, lr=0.1 |
| GradBoost | sklearn | n_estimators=100, max_depth=4, lr=0.1 |
| AdaBoost | sklearn | n_estimators=100, lr=0.1 |
| KNN | sklearn | n_neighbors=10, n_jobs=-1 |
| LogReg | sklearn | max_iter=1000 |
| ElasticNet | sklearn | LogisticRegression, penalty=elasticnet, solver=saga, l1_ratio=0.5 |
| SVM | sklearn | SVC, kernel=rbf, probability=True |
| Dummy | sklearn | DummyClassifier, strategy=prior |

*Table 4.* Foundation model and RDBLearn hyperparameters. Training set capped at 50,000 samples. Timeout: 900s (FMs), 7,200s (RDBLearn).

| Model | Source | Key Settings |
|---|---|---|
| TabPFN | tabpfn | Pre-trained; flat-table features via simple joins; 50K sample cap |
| TabICL | tabicl | Pre-trained; flat-table features; 50K sample cap |
| ContextTab | contexttab | Pre-trained; flat-table features; 50K sample cap |
| TabSTAR | tabstar | Pre-trained; flat-table features; 50K sample cap |
| RealMLP | realmlp | Default hyperparameters; flat-table features |
| RDB+TabPFN | rdblearn | Deep Feature Synthesis (fastdfs) + TabPFN |
| RDB+TabICL | rdblearn | Deep Feature Synthesis + TabICL |
| RDB+CtxTab | rdblearn | Deep Feature Synthesis + ContextTab |

*Table 5.* Graph model hyperparameters. All trained on the relational graph via relbench.modeling.

| Model | Architecture | Key Settings |
|---|---|---|
| GNN | HeteroGraphSAGE | RelBench reference implementation; 5,400s timeout |

*Table 6.* Full G-theory variance decomposition for all 14 benchmarks. $\bar{n}_I$: average items per task. $\sigma^2$ columns show raw variance components. SEM: standard error of measurement for model means.

| Dataset | $n_M$ | $n_T$ | $\bar{n}_I$ | $\sigma_M^2$ | $\sigma_T^2$ | $\sigma_{MT}^2$ | $\sigma_e^2$ | Spread | %M | %T | %MT | %e | $E\rho^2$ | $\Phi$ |
|---|---|---|---|---|---|---|---|---|---|---|---|---|---|---|
| dbi-amazon | 32 | 2 | 10000 | 0.0000 | 0.0096 | 0.0043 | 0.0000 | 0.260 | 0.0 | 68.9 | 31.0 | 0.1 | 0.000 | 0.000 |
| dbi-avs | 32 | 1 | 10000 | 0.0000 | 0.0000 | 0.0000 | 0.0000 | 0.013 | 21.0 | 0.0 | 0.0 | 79.0 | 0.210 | 0.210 |
| dbi-diginetica | 29 | 1 | 6616 | 0.0001 | 0.0000 | 0.0000 | 0.0000 | 0.031 | 95.3 | 0.0 | 0.0 | 4.7 | 0.953 | 0.953 |
| dbi-seznam | 29 | 2 | 10000 | 0.0579 | 0.0003 | 0.0025 | 0.0000 | 0.821 | 95.3 | 0.5 | 4.2 | 0.0 | 0.979 | 0.976 |
| dbi-stackex. | 29 | 2 | 10000 | 0.0001 | 0.0071 | 0.0010 | 0.0000 | 0.192 | 1.1 | 86.9 | 12.0 | 0.0 | 0.150 | 0.021 |
| rel-amazon | 31 | 5 | 10000 | 0.0001 | 0.0078 | 0.0007 | 0.0000 | 0.249 | 1.0 | 91.2 | 7.8 | 0.0 | 0.394 | 0.049 |
| rel-arxiv | 35 | 3 | 10000 | 0.0005 | 0.0186 | 0.0032 | 0.0000 | 0.329 | 2.2 | 83.2 | 14.5 | 0.1 | 0.315 | 0.064 |
| rel-avito | 34 | 3 | 7272 | 0.0000 | 0.0047 | 0.0059 | 0.0000 | 0.260 | 0.0 | 44.5 | 55.4 | 0.1 | 0.000 | 0.000 |
| rel-event | 35 | 3 | 1387 | 0.0002 | 0.0102 | 0.0025 | 0.0001 | 0.314 | 1.4 | 79.0 | 19.1 | 0.5 | 0.176 | 0.041 |
| rel-f1 | 35 | 3 | 729 | 0.0000 | 0.0010 | 0.0000 | 0.0000 | 0.074 | 0.3 | 98.6 | 0.8 | 0.4 | 0.393 | 0.008 |
| rel-hm | 31 | 3 | 10000 | 0.0001 | 0.0058 | 0.0007 | 0.0000 | 0.264 | 1.6 | 87.3 | 11.1 | 0.0 | 0.302 | 0.047 |
| rel-ratebeer | 33 | 5 | 9757 | 0.0002 | 0.0074 | 0.0015 | 0.0000 | 0.211 | 1.8 | 81.8 | 16.4 | 0.1 | 0.355 | 0.084 |
| rel-salt | 34 | 8 | 10000 | 0.0041 | 0.0201 | 0.0088 | 0.0000 | 0.561 | 12.4 | 61.0 | 26.6 | 0.0 | 0.788 | 0.531 |
| rel-stack | 34 | 3 | 10000 | 0.0000 | 0.0010 | 0.0044 | 0.0000 | 0.315 | 0.3 | 18.4 | 81.2 | 0.0 | 0.011 | 0.009 |

*Table 7.* D-study item subsampling: mean Spearman $\rho$ (std) at each item fraction. Values $\geq 0.90$ in bold.

| Dataset | 1% | 2% | 5% | 10% | 20% | 30% | 50% | 70% | 100% |
|---|---|---|---|---|---|---|---|---|---|
| rel-salt | **.924** | **.943** | **.951** | **.971** | **.979** | **.985** | **.989** | **.994** | 1.00 |
| rel-event | .813 | .857 | **.916** | **.942** | **.957** | **.971** | **.985** | **.989** | 1.00 |
| rel-ratebeer | .770 | .845 | **.904** | **.924** | **.943** | **.953** | **.965** | **.970** | 1.00 |
| rel-arxiv | .655 | .771 | .832 | **.907** | **.943** | **.958** | **.981** | **.990** | 1.00 |
| rel-amazon | .661 | .741 | .813 | .878 | **.913** | **.935** | **.956** | **.973** | 1.00 |
| rel-stack | .644 | .747 | .826 | .854 | **.911** | **.928** | **.959** | **.978** | 1.00 |
| dbi-amazon | .714 | .793 | .887 | **.927** | **.956** | **.972** | **.984** | **.990** | 1.00 |
| dbi-seznam | .537 | .638 | .713 | .822 | .895 | **.927** | **.957** | **.978** | 1.00 |
| rel-hm | .584 | .663 | .752 | .822 | .839 | .873 | .891 | **.917** | 1.00 |
| rel-f1 | .201 | .292 | .411 | .597 | .689 | .779 | .882 | **.940** | 1.00 |
| rel-avito | .424 | .451 | .522 | .643 | .753 | .786 | .896 | **.941** | 1.00 |
| dbi-stackex. | .541 | .547 | .572 | .570 | .613 | .643 | .612 | .674 | 1.00 |
| dbi-diginetica | .135 | .193 | .169 | .150 | .160 | .191 | .253 | .267 | 1.00 |
| dbi-avs | .032 | .057 | .082 | .081 | .176 | .132 | .179 | .234 | 1.00 |

*Table 8.* D-study task subsampling: mean Spearman $\rho$ at each number of tasks $k$. Values $\geq 0.90$ in bold.

| Dataset | $k{=}1$ | $k{=}2$ | $k{=}3$ | $k{=}4$ | $k{=}5$ | $k{=}6$ | $k{=}7$ | $k{=}8$ |
|---|---|---|---|---|---|---|---|---|
| rel-salt | .331 | .541 | .780 | .836 | .895 | **.962** | **.955** | 1.00 |
| rel-amazon | .409 | .609 | .733 | .877 | 1.00 | | | |
| rel-ratebeer | .347 | .593 | .784 | **.924** | 1.00 | | | |
| rel-f1 | .641 | **.913** | 1.00 | | | | | |
| rel-arxiv | .479 | .831 | 1.00 | | | | | |
| rel-stack | .749 | .872 | 1.00 | | | | | |
| rel-event | .554 | .808 | 1.00 | | | | | |
| rel-hm | .463 | .820 | 1.00 | | | | | |
| rel-avito | .389 | .697 | 1.00 | | | | | |