# OpenReview forum: "Are Tabular Foundation Model Rankings Reliable? A Generalizability Theory Analysis of RelBench and DBInfer"
_ICML.cc/2026/Workshop/FMSD — FMSD @ ICML 2026 Poster_

### Official Review · Reviewer_BD5B · 2026-05-16

**Rating:** 7
**Confidence:** 4

**Review:**

Summary of contributions

This paper investigates the reliability of leaderboard rankings for tabular and relational foundation models using Generalizability Theory (G-theory). The authors analyze 14 datasets from RelBench and DBInfer, covering 48 tasks and up to 35 models, and decompose performance variance into model effects, task effects, model×task interaction, and sampling noise. The main finding is that most benchmark rankings are unreliable: only a small fraction of datasets achieve high generalizability (Eρ² > 0.80), while model×task interaction dominates variance in many cases.

Strengths
1. Strong and timely problem formulation. The paper addresses an important yet underexplored issue: whether widely used tabular and relational foundation model leaderboards actually provide reliable rankings. This is highly relevant given the rapid development of tabular FMs.
2. Solid theoretical grounding in G-theory. The use of Generalizability Theory provides a principled and interpretable framework for decomposing variance sources. Compared to ad-hoc ranking stability analysis, this offers a more rigorous statistical foundation for evaluating benchmark reliability.
3. Comprehensive empirical analysis. The study spans multiple datasets (RelBench and DBInfer), a large number of tasks, and diverse model families. The inclusion of D-study simulations (item and task subsampling) provides additional insight into benchmark redundancy and efficiency.

Weaknesses
1. Strong assumptions about task exchangeability. The analysis treats tasks as exchangeable random effects, which may not fully hold in practice since tasks in RelBench and DBInfer are often intentionally heterogeneous and semantically structured. This may lead to overestimation of model×task interaction variance.
2. Limited connection between variance decomposition and actionable modeling insights. While the paper clearly shows that rankings are unstable, it is less clear how this should concretely influence model design or training strategies. The analysis is diagnostic rather than prescriptive.
3. Potential sensitivity to model selection and evaluation protocol. The variance estimates may depend on the specific set of models included (e.g., strong imbalance between classical models and foundation models). It is unclear how robust the conclusions are under alternative model sets or future stronger baselines.

Suggestions
1. It would strengthen the paper to include sensitivity analysis over different task groupings (e.g., grouping by domain/type) to better understand whether instability persists at coarser semantic levels.
2. The authors could provide more practical guidance for benchmark design, such as recommended numbers of tasks, or structured sampling strategies that reduce model×task interaction effects.
3. The paper would benefit from improved figure readability. In particular, some annotations and labels in Figure 1 appear blurry or difficult to read, which makes the visualization harder to interpret.

---

### Official Review · Reviewer_G3de · 2026-05-20

**Rating:** 6
**Confidence:** 4

**Review:**

Summary:
This paper applies generalizability theory to quantify ranking reliabilities of up to 35 models from 11 families across 14 datasets. Experimental findings suggest that single number leaderboard rankings of tabular and relational learning foundation models may provide an unstable estimate of relative model performance.

Strengths:
1. This paper proposes a novel and interesting angle which analyzing the ranking reliabilities of tabular and relational learning foundation models.
2. Adequate experiments have been conducted.
3. Proving a good angle foe AI users when they choosing which foundation model to be used.

Weaknesses:
1. Is G-theory alone could adequately and comprehensively analysis the ranking reliabilities remains questionable.

---

### Official Review · Reviewer_q5cw · 2026-05-21
**A valuable analysis of tabular foundation model ranking reliability, though an important benchmark is missing**

**Rating:** 7
**Confidence:** 4

**Review:**

### **Summary**

The paper studies the reliability of model rankings in tabular and relational benchmarks. It applies Generalizability Theory to decompose sources of variance in benchmark scores, including model effects, task effects, model-task interactions, and sampling error. The analysis covers RelBench and DBInfer, with 14 datasets, 48 tasks, and several model families. The main finding is that only 2 of the 14 datasets achieve reliable rankings under the usual generalizability threshold, suggesting that single-number leaderboard rankings may often be unstable.

### **Strengths**

The paper is generally well written and easy to follow.

The use of Generalizability Theory is well motivated and provides a principled framework for separating model signal from benchmark-induced noise.

Assessing the reliability of model rankings is an important and valuable contribution, especially as tabular and relational foundation models are increasingly compared through aggregate benchmark scores.

The experiments are well conducted and sufficiently detailed.

### **Areas for Improvement**

My main concern is the choice of benchmark. It is unclear why the authors did not perform the analysis on TabArena, which is now a standard live leaderboard for tabular models. This would have made the significance of the findings easier to assess, since TabArena already provides established rankings across a broad set of models, including tabular foundation models.

The paper also does not sufficiently leverage the D-study analysis to provide concrete recommendations on how to increase the reliability of current benchmarks. The current results indicate that some benchmarks contain redundant test items and that task diversity is important, but the paper could be more prescriptive about how many tasks or items are needed in practice.

The paper could better explain how its conclusions relate to benchmark use by practitioners. If rankings are unreliable, should users avoid aggregate rankings altogether, report confidence intervals, or use per-task rankings? Clearer guidance would strengthen the impact.

### **Detailed Comments**

The authors should justify the choice of RelBench and DBInfer more clearly and explain why TabArena was not included. Even if TabArena is outside the paper’s main scope, a discussion of how the proposed analysis could apply to it would improve the paper.

The D-study results are interesting, but the paper should provide more concrete benchmark-design recommendations. For example, it could specify when adding tasks is more valuable than adding test examples, and how benchmark designers should decide whether a dataset has enough tasks.

It would be useful to report reliability coefficients alongside leaderboard results. This could make the analysis more transparent.

The paper should clarify whether low ranking reliability necessarily means that the benchmark is poor, or whether it means that models have genuinely task-specific behavior. This distinction is important for interpreting the results fairly.

### **Justification of Score**

The paper addresses an important and timely problem: whether benchmark rankings for tabular and relational models are reliable. The methodology is appropriate, the paper is clearly written, and the empirical analysis provides useful evidence that aggregate leaderboard rankings can be unstable. My main reservations concern the benchmark choice and the need for more actionable recommendations from the D-study analysis. Overall, this is a solid and relevant contribution to the workshop.